# Integrated Process for Producing Glycolic Acid from Carbon Dioxide Capture Coupling Green Hydrogen

**Dongliang Wang** [1,2,*] **, Jingwei Li** [1]**, Wenliang Meng** [1]**, Jian Wang** [1]**, Ke Wang** [1]**, Huairong Zhou** [1,2]**, Yong Yang** [1,2]**, Zongliang Fan** [1,2] **and Xueying Fan** [3]

1 School of Petrochemical Engineering, Lanzhou University of Technology, Lanzhou 730050, China
2 Key Laboratory of Low Carbon Energy and Chemical Engineering of Gansu Province, Lanzhou 730050, China
3 Automation Institute of Lanzhou Petrochemical Company, Lanzhou 730050, China
* Correspondence: wangdl@lut.edu.cn

**Abstract:** A novel process path is proposed to produce glycolic acid (GA) from $CO_2$ as the feedstock, including $CO_2$ capture, power-to-hydrogen, $CO_2$ hydrogenation to methanol, methanol oxidation to formaldehyde, and formaldehyde carbonylation units. The bottlenecks are discussed from the perspectives of carbon utilization, $CO_2$ emissions, total site energy integration, and techno-economic analysis. The carbon utilization ratio of the process is 82.5%, and the $CO_2$ capture unit has the largest percentage of discharge in carbon utilization. Among the indirect emissions of each unit, the $CO_2$ hydrogenation to methanol has the largest proportion of indirect carbon emissions, followed by the formaldehyde carbonylation to glycolic acid and the $CO_2$ capture. After total site energy integration, the utility consumption is 1102.89 MW for cold utility, 409.67 MW for heat utility, and 45.98 MW for power. The $CO_2$ hydrogenation to methanol makes the largest contribution to utility consumption due to the multi-stage compression of raw hydrogen and the distillation of crude methanol. The unit production cost is 834.75 $/t-GA; $CO_2$ hydrogenation to methanol accounts for the largest proportion, at 70.8% of the total production cost. The total production cost of the unit depends on the price of hydrogen due to the currently high renewable energy cost. This study focuses on the capture and conversion of $CO_2$ emitted from coal-fired power plants, which provides a path to a feasible low-carbon and clean use of $CO_2$ resources.

**Keywords:** $CO_2$ capture; renewable hydrogen; glycolic acid synthesis; process modeling; process analysis





## 1. Introduction

Polyglycolic acid (PGA) is the simplest linear aliphatic polyester in terms of chemical structure [1]. Compared to polyhydroxyalkanoates such as polylactic acid and poly (3-hydroxybutyric acid, PGA has strong intermolecular hydrogen-bond interaction, a highly regular molecular structure, and densely packed molecular chains, with characteristics such as high crystallinity, high density, and brilliant heat resistance [2]. PGA also exhibits great mechanical properties [3], as well as unique biodegradability, good biocompatibility, and a prominent gas barrier [4]. PGA is one of the most widely studied and applied biodegradable materials [5].

PGA is mainly produced by direct polycondensation of glycolic acid or ring-opening polymerization of glycolide (i.e., glycolic acid is first cyclized and polymerized into ethylene cross-ester, which is then ring-opened and polymerized into PGA). At present, there are four main production methods for the preparation of glycolic acid, namely, the hydrolysis of chloroacetic acid [6], cyanidation [7], electroreduction of oxalic acid to glycolic acid [8], and formaldehyde carbonylation [9]. Chloroacetic acid hydrolysis is the most traditional preparation method, with a long process and low yield, and is only suitable for small-scale production. Although the cyanidation method has the advantage of steady, handy operation and high product purity, the safety requirements for production operations are

high due to the presence of highly toxic cyanide. Electroreduction of oxalic acid to glycolic acid has certain requirements on the flow rate, current density, and voltage of electrolyte, and the energy consumption of electroreduction is high. Formaldehyde carbonylation is a typical Koch reaction over acid catalyst, in which formaldehyde, carbon monoxide, and water are used as raw materials to synthesize glycolic acid [10]. If solid acid catalysts are developed with good catalytic performance, the carbonylation reaction of formaldehyde could be carried out at low pressure, with the advantages of low-cost production of raw materials and simple separation [11]; this has good development prospects and has thus been widely considered. Formaldehyde, the raw material for formaldehyde carbonylation, is mainly produced by the oxidation of methanol. As the most important greenhouse gas in the atmosphere, $CO_2$ is responsible for more than 60% of the warming effect [12], while methanol is an important platform product to realize $CO_2$ resource utilization [13].

In this study, a process for transforming $CO_2$ to glycolic acid is proposed, starting from $CO_2$ capture, including $CO_2$ hydrogenation to methanol, methanol oxidation to formaldehyde, and formaldehyde carbonylation. Firstly, a detailed process model is prepared, and the material and energy consumption of the whole system are analyzed. The bottlenecks are discussed from the perspectives of carbon utilization, energy analysis and integration, and $CO_2$ emissions. Based on the energy flow data, a detailed energy analysis of each unit is carried out using the pinch technique, and total site composite curves are constructed to analyze the maximum energy saving potential in the context of plant-wide energy integration. Finally, the energy efficiency and economic indicators of the integration scheme are assessed.

The rest of this paper is organized as follows. The reaction mechanism and process for transforming $CO_2$ to glycolic acid are described in Section 2. Furthermore, the parameter optimization of each unit is discussed, to obtain the optimal simulation results. Process system heat integration to improve the energy efficiency is then considered in Section 3. The material balance results are introduced, and the carbon utilization, $CO_2$ emissions, and techno-economic analysis are presented. The conclusions are drawn in Section 5.

## 2. Procedure Description and Parameter Optimization

The process proposed in this study for transforming carbon dioxide into glycolic acid based on hydrogen production from renewable energy sources is shown in Figure 1. It mainly consists of five units: electrolytic water to hydrogen, carbon dioxide capture, carbon dioxide hydrogenation to methanol, methanol oxidation to formaldehyde, and formaldehyde carbonylation to glycolic acid.

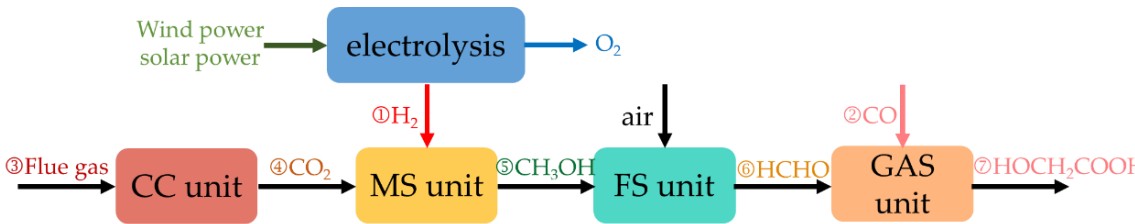

**Figure 1.** Carbon dioxide to polyglycolic acid process.

The flue gas emitted from coal-fired power plants is captured and purified through carbon capture (CC unit) to obtain high-purity carbon dioxide, which is then synthesized through methanol synthesis (MS unit) with hydrogen produced by electrolytic water to get methanol, followed by oxidation with oxygen to gain formaldehyde via formaldehyde synthesis (FS unit), and finally, with carbon monoxide and water in the presence of a catalyst, glycolic acid synthesis (GAS unit) to obtain the end product of glycolic acid.

Proton exchange membrane water electrolysis is selected for the transformation of electrolytic water to hydrogen, the phase change solvent is used for $CO_2$ capture, and direct $CO_2$ hydrogenation to methanol with a Cu/Zn/Al/Zr catalyst is selected for the MS unit. Meanwhile, formaldehyde production over iron-molybdenum oxide as the catalyst is used

for the FS unit, and a metal solid acid catalyst is used for formaldehyde carbonylation to glycolic acid in this study.

### 2.1. Electrolytic Water Hydrogen Production

Water electrolysis has benefits over other traditional hydrogen-generation technologies associated with fossil energy due to the reduced carbon emissions when it is integrated with a renewable source of energy. At the technical level, water electrolysis is mainly divided into alkaline water electrolysis (AWE) [14], proton exchange membrane (PEM) water electrolysis [15], solid polymer anion exchange membrane water electrolysis [16], and solid oxide water electrolysis [17]. Among them, both AWE and PEM have been widely used for large-scale industrial applications. Specifically, PEM water electrolysis is a promising technology for hydrogen-generation applications, with good chemical stability, proton conductivity, and electrolytic pollution-free corrosion. The intermittency and volatility of wind and solar power are major obstacles to generating large amounts of electricity from renewable sources. Yet, PEM water electrolysis hydrogen production technology can adapt to the intermittency and fluctuation of renewable energy generation. The electrolytic water hydrogen production process is shown in Figure 2, which first vaporizes the desalinated water and then mixes it with the back-loop cooling products. The mixed material is further heated to 100 °C, and then it is partially decomposed into hydrogen and oxygen in the electrolyzer. The generated gas is cooled to 40 °C and enters the flash evaporator, where most of the unreacted water is separated. The hydrogen (99.9 vol.%) is fed into the methanol synthesis unit through the gas treatment system. The key parameters of the electrolyzer are shown in Table 1.

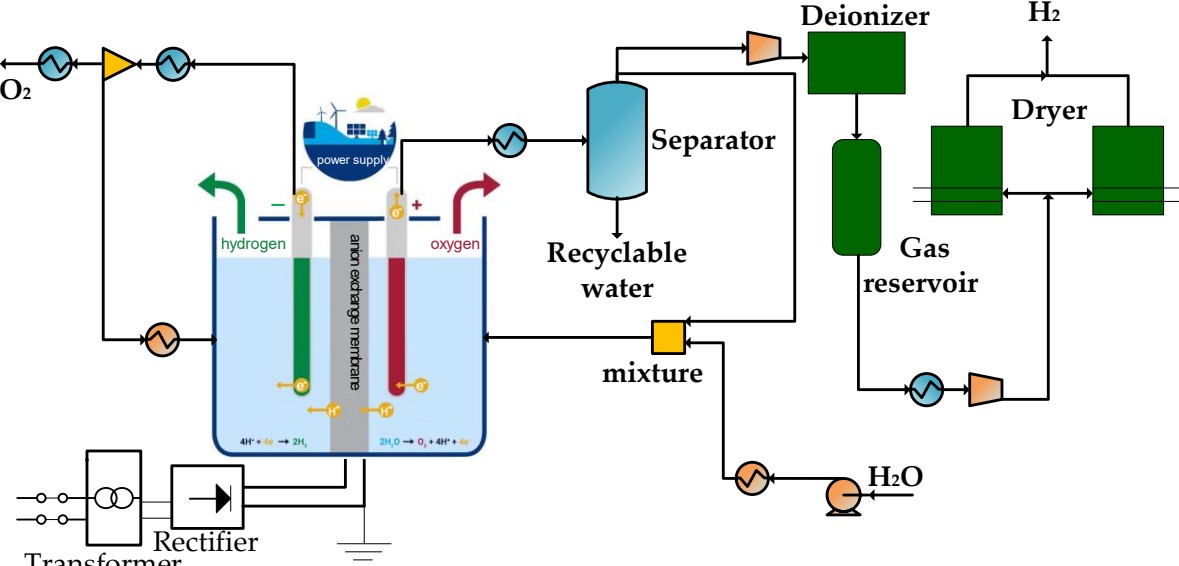

**Figure 2.** Electrolytic water hydrogen production.

According to electrochemical principles, the rate of hydrogen production from water electrolysis is proportional to the current magnitude. Since there is a loss of current during hydrogen production by water electrolysis, this reduces hydrogen production. Hence, the actual hydrogen production rate of the electrolyzer can be expressed as [18]:

$$q_{H_2} = 1800\eta_F n_c \frac{I_{elec}}{F} \tag{1}$$

where $q_{H2}$ is the hydrogen production rate, mol/h, $\eta_F$ is the current density of electrolytic cells, $F$ is Faraday's constant, C/mol, $I_{elec}$ is the current value, and $n_c$ is the number of electrolytic cell arrays.

**Table 1.** Key technical parameters of the electrolysis [19].

| Parameter | Value |
|---|---|
| power consumption production | 4.50 kWh/Nm$^3$ H$_2$ |
| unit cost of hydrogen production | 3.13 \$/kg H$_2$ |
| hydrogen purity | 99.99% |
| working pressure (bar) | 30.00 |
| energy efficiency | 80.00% |

*2.2. CO$_2$ Capture by Phase Change Absorbent (CC Unit)*

2.2.1. Reaction Mechanism of CO$_2$ Capture by Phase Change Absorbent

CO$_2$ capture technologies mainly include physical adsorption [20], chemical absorption [21], membrane separation [22], low-temperature distillation [23], etc. Among carbon capture technologies, the most mature applied technology for CO$_2$ capture is chemical absorption/desorption using an aqueous amine solution of monoethanolamine (MEA) [24]. The reaction of MEA and CO$_2$ consists of the following instantaneous reactions:

$$H_2O \leftrightarrow H_3O^+ + OH^- \tag{2}$$

$$CO_2 + 2H_2O \leftrightarrow HCO_3^- + H_3O^+ \tag{3}$$

$$HCO_3^- + H_2O \leftrightarrow CO_3^{2-} + H_3O \tag{4}$$

$$MEACOO^- + H_2O \leftrightarrow MEA + HCO_3^- \tag{5}$$

$$MEAH^+ + H_2O \leftrightarrow MEA + H_3O^+ \tag{6}$$

The reaction models of the absorber/stripper use power-law expressions and the following finite rate reactions, with the parameters required by the expressions shown in Table 2.

$$CO_2 + OH^- \rightarrow HCO_3^- \tag{7}$$

$$HCO_3^- \rightarrow CO_2 + OH^- \tag{8}$$

$$MEA + CO_2 + H_2O \rightarrow MEACOO^- + H_3O^+ \tag{9}$$

$$MEACOO^- + H_3O^+ \rightarrow MEA + CO_2 + H_2O \tag{10}$$

$$r = A_i exp(\frac{-E_i}{RT})\prod_{i=1}^{N}(\chi_i\gamma_i)^{\alpha_i} \tag{11}$$

where $r$ is the reaction rate, $A_i$ is the pre-exponential factor, $T$ is the absolute temperature, K, $E_i$ is the activation energy (kcal/mol), $R$ is the universal gas constant (cal/(mol K)), $\chi_i$ is the mole fraction of the component, $\gamma_i$ is the activity coefficient of components in the reaction equation, $\alpha_i$ is the stoichiometric coefficient of component $i$ in the reaction, and $N$ is the number of components.

**Table 2.** Reaction kinetic parameters $A_i$ and $E_i$ in finite rate reactions.

| Reaction | Pre-Exponential Factor $A_i$ | Activation Energy $E_i$ (kcal/mol) |
|---|---|---|
| 7 | $1.33 \times 10^{17}$ | 13.249 |
| 8 | $6.63 \times 10^{16}$ | 25.656 |
| 9 | $3.02 \times 10^{14}$ | 9.8558 |
| 10 (absorber) | $5.52 \times 10^{23}$ | 16.518 |
| 10 (stripper) | $6.50 \times 10^{27}$ | 22.782 |

The high energy consumption of the MEA absorbent absorption process is still a major obstacle to further development [25]. To improve the cyclic capacity and reduce the energy consumption of desorption, Zheng et al. proposed a new CO$_2$ absorbent, the phase change absorbent [26]. From this, a phase change absorbent based on MEA was proposed [27],

where the desorption energy consumption can be reduced from 3.99 GJ/t-$CO_2$ of MEA absorbent to 2.4 GJ/t-$CO_2$. The conceptual model is shown in Figure 3.

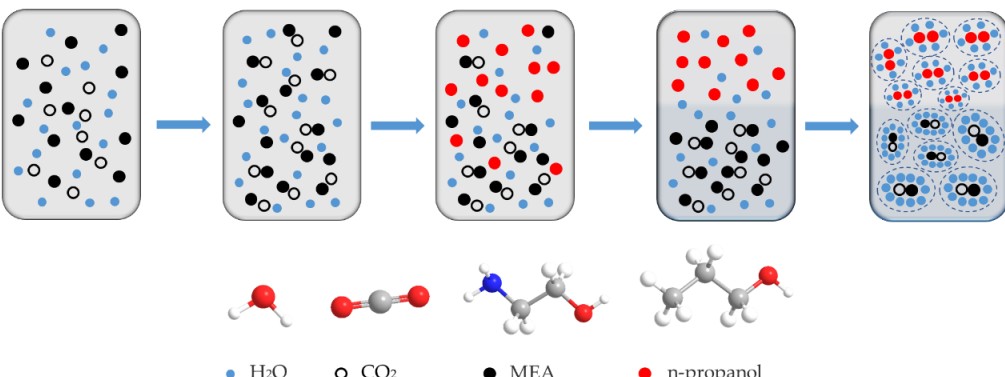

● $H_2O$    ○ $CO_2$    ● MEA    ● n-propanol

**Figure 3.** $CO_2$ phase change capture conceptual model.

When 1-propanol is added to the aqueous solution of MEA, as the $CO_2$ absorption reaction proceeds, ionic species, carbamate, and protonated MEA are formed, which could be regarded as MEA salt. With the addition of salt, phase separation occurs for a water-miscible organic liquid from its aqueous solution, known as the salt-out effect. As the initial 1-propanol concentration increases, the water concentration decreases, and the concentration of MEA salt on the water base increases after $CO_2$ absorption. More water clusters gradually evolve around the MEA salt, and the n-propanol molecules are excluded by the water and form clusters. When the MEA salt reaches a certain concentration, the number of water molecules that evolve around the MEA salt increases, and in this case, the water available to interact with 1-propanol decreases. As a result, more 1-propanol is forced to the upper phase, which causes the 1-propanol concentration to increase in the upper phase and decrease in the lower phase, as shown in Figure 3.

In comparison to conventional aqueous solutions, the lower phase solution, which is rich in carbamate and protonated MEA, enters the desorption unit. The potential advantages of the phase change absorbent include not only the higher $CO_2$ cyclic capacity but also the lower liquid flow rate in the stripper. Herein, the regeneration energy consumption will be significantly reduced. The separation of the two phases by decantation can reduce the solvent mass flow rate at the stripper; meanwhile, the richer $CO_2$ loading can lower the energy consumption of sensible heat and water vaporization heat. According to the volume fraction of the upper phase and $CO_2$ distribution in biphasic solvent (illustrated in Figure S1), a phase change absorbent of MEA/n-propanol/water with a mass ratio of 3/3/4 is chosen for the unit [28].

### 2.2.2. Procedure for $CO_2$ Capture by Phase Change Absorbent

The flue gas composition in this study is shown in Table 3, based on data provided by the National Energy Laboratory for a 550 MW coal-fired power plant. The operating conditions of the absorber in this unit are 1.1 bar and 40–60 °C. The heat loss and pressure loss in the absorber are negligible, and the operating conditions of the stripper are 1.1 bar and 90–130 °C. The pressure loss in the stripper is also negligible. The cooler is used to ensure that the temperature of the solution entering the absorber tower is 40 °C, and the heat exchanger is used to maximize the heat exchange between the hot and cold streams.

During the $CO_2$ absorption as shown in Figure 4, the flue gas from the coal-fired power plant is washed by the water washing tower (removing solid impurities and sulfide in the flue gas), then the flue gas temperature is controlled at about 40 °C. The flue gas enters the absorber from the bottom of the column, and the lean phase enters the absorber from the top of the column, so the two streams are in counter-current contact. The net flue gas after decarbonization is discharged from the top of the absorber tower, and the rich phase is discharged from the bottom of the absorber. The rich phase forms two phases in the phase

separator: the upper phase is the $CO_2$-lean phase, and the lower phase is the $CO_2$-rich phase. The $CO_2$-rich phase is pressurized by the pump and then enters the heat exchanger, then sprays into the top of the stripper, where $CO_2$ in the $CO_2$-rich phase is desorbed by the heat provided by the reboiler at the bottom of the stripper, before the desorbed $CO_2$ is separated by condensation at the top of the stripper. The $CO_2$-lean phase discharged from the bottom of the stripper is cooled by the heat exchange with the $CO_2$-rich phase and then reenters the absorber for recycling. A comparison of the experimental density and model density of the MEA/1-propanol system with different $CO_2$ loading levels (at 293 K) is shown in Table S1.

**Table 3.** Composition of the flue gas.

| Working Condition and Composition | Value |
|---|---|
| $N_2$ (mol.%) | 77.90 |
| $CO_2$ (mol.%) | 14.60 |
| $O_2$ (mol.%) | 3.30 |
| $H_2O$ (mol.%) | 4.20 |
| temperature (°C) | 42.00 |
| Pressure (kPa) | 101.33 |
| molar flow rate (kmol/h) | 40,000.00 |

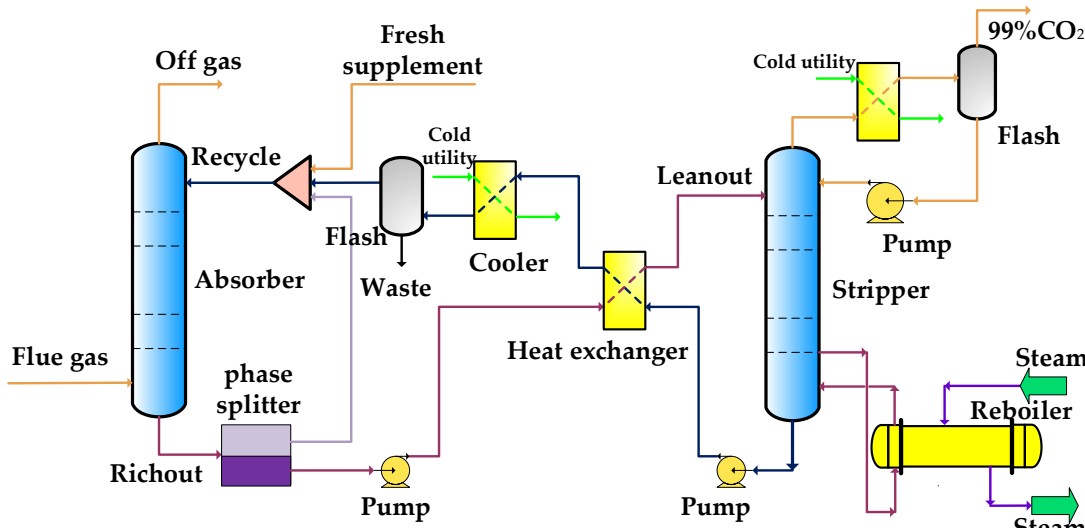

**Figure 4.** $CO_2$ capture based on a phase change absorbent.

2.2.3. Parameter Optimization for $CO_2$ Capture by Phase Change Absorbent

According to the recommended requirements of the U.S. Department of Energy (DOE) for $CO_2$ capture, there should be a 95% $CO_2$ recovery purity and 90% capture rate [29]. Therefore, this study mainly analyzes and optimizes the factors influencing the $CO_2$ purity and $CO_2$ capture rate. The corresponding equations are shown as follows:

$$\alpha = \frac{M_2}{M_1} \times 100\% \tag{12}$$

$$\psi = \frac{M_2}{M_3} \times 100\% \tag{13}$$

where $\alpha$ is the purity; $\psi$ is the desorption rate; $M_1$ is the mass flow rate of product gas at the top of the stripper, kg/h; $M_2$ is the mass flow rate of $CO_2$ contained in the product gas at the top of the stripper, kg/h; $M_3$ is the mass flow rate of $CO_2$ contained in the flue gas, kg/h.

The reflux ratio and reboiler duty are the key parameters affecting the $CO_2$ purity and capture rate in the stripper. The relationship between the impacts of the reflux ratio and reboiler duty on the $CO_2$ purity and desorption rate is shown in Figure 5.

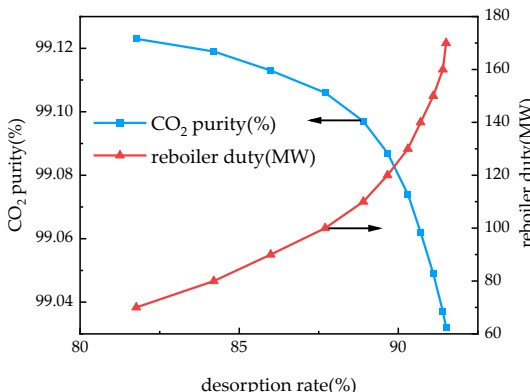

**Figure 5.** Impact of reboiler duty on $CO_2$ purity and desorption rate.

As the $CO_2$ desorption rate increases, the $CO_2$ purity in desorption gas decreases, especially when the $CO_2$ desorption rate exceeds 90%, and the $CO_2$ purity in desorption gas decreases significantly. For example, when the desorption rate increases from 90% to 90.5%, the $CO_2$ purity in desorption gas decreases by about 0.1%. Accordingly, the $CO_2$ purity in desorption gas decreases by about 0.2% when the desorption rate increases from 90.5% to 91%. In industrial production, the $CO_2$ desorption rate should be weighed against the $CO_2$ purity in the desorption gas. The desorption rate is also the main factor affecting the reboiler duty. As the desorption rate increases, the reboiler duty of the desorption column increases, especially when the desorption rate exceeds 90%, and the reboiler duty rises apparently. Therefore, the $CO_2$ desorption rate is specified as 90% in this study, while the calculated desorption energy consumption is 150 MW, equivalent to a unit energy consumption of 2.42 GJ/t-$CO_2$. The flow rate results for the carbon capture are shown in the supporting information (illustrated in Table S2).

*2.3. $CO_2$ Hydrogenation to Methanol (MS Unit)*

2.3.1. Reaction Mechanism for $CO_2$ Hydrogenation to Methanol

The $CO_2$ to methanol process is categorized as direct $CO_2$ hydrogenation to methanol [30] and $CO_2$ hydrogenation to form methanol via a reverse-water-gas-shift reaction (CAMERE) [31]. The direct $CO_2$ hydrogenation to methanol process uses $CO_2$ and $H_2$ as the feedstock and synthesizes methanol directly through a catalytic reaction, which is also known as the direct or one-step process. The CAMERE process also uses $CO_2$ and $H_2$ as the feedstock and converts $CO_2$ to syngas via a reverse-water-gas shift (RWGS) reaction, and then synthesizes methanol; this technology is also known as the indirect or two-step process. The direct process has lower fixed cost investment, lower product costs, and simpler process control than the indirect process.

The main reactions involved in the hydrogenation of carbon dioxide to methanol include:

$$CO_2 + 3H_2 \rightarrow CH_3OH + H_2O \tag{14}$$

$$CO_2 + H_2 \rightarrow CO + H_2O \tag{15}$$

$$CO + 2H_2 \rightarrow CH_3OH \tag{16}$$

The influence of temperature and pressure on the equilibrium composition has a thermodynamic root cause, which can be analyzed by the influence of reactions (14)–(16). Lowering the temperature promotes reactions (14) and (16) to proceed in the positive direction, and the conversion rate of $CO_2$ increases. At the same time, due to the exothermicity of the main reaction (14) or (16) and endothermic characteristics for reaction (15), a low

temperature is favorable to methanol formation, and the selectivity of methanol increases when the temperature decreases. But from a dynamic perspective, a low temperature will reduce the catalytic reaction rate. An increasing pressure favors the reaction proceeding in the positive direction because the number of molecules decreases in the $CO_2$ hydrogenation primary reaction. However, high pressure increases the compression power consumption and the operation cost.

The Langmuir–Hinshelwood (LHHW) model is chosen for the reaction kinetics [32], and the reaction equilibrium constants are based on those used in the study of Lim et al. [33] and the experimental data of Graaf et al. [34]. The $CO_2$ hydrogenation to methanol is based on the Redlich–Kwong–Soave thermodynamic model. The kinetic rate expressions are as follows for reactions (14)–(16) and the parameters required by the expressions are shown in Tables S3–S5. Fugacity (*f*) is used because of the high pressure in the reactions.

$$r_{CH_3OH} = k_1 \frac{K_{CO}\left[f_{CO}f_{H_2}^{3/2} - f_{CH_3OH}/K_A\sqrt{f_{H_2}}\right]}{(1 + K_{CO}f_{CO} + K_{CO_2}f_{CO_2})\left[\sqrt{f_{H_2}} + (K_{H_2O}/\sqrt{K_H})f_{H_2O}\right]} \tag{17}$$

$$r_{CO} = k_2 \frac{K_{CO_2}\left[f_{CO_2}f_{H_2} - f_{H_2O}f_{CO}/K_B\right]}{(1 + K_{CO}f_{CO} + K_{CO_2}f_{CO_2})\left[\sqrt{f_{H_2}} + (K_{H_2O}/\sqrt{K_H})f_{H_2O}\right]} \tag{18}$$

$$r_{CH_3OH} = k_3 \frac{K_{CO_2}\left[f_{CO_2}f_{H_2}^{3/2} - f_{H_2O}f_{CH_3OH}/\left(f_{H_2}^{3/2}K_C\right)\right]}{(1 + K_{CO}f_{CO} + K_{CO_2}f_{CO_2})\left[\sqrt{f_{H_2}} + (K_{H_2O}/\sqrt{K_H})f_{H_2O}\right]} \tag{19}$$

The comparison between the experimental data and simulation results (illustrated in Table 4) shows good agreement, with less than a 5% error around the operating process conditions, indicating correct implementation of the kinetics.

**Table 4.** Comparison of experimental data with simulation results.

| | **CO$_2$ Conversion** | | **Methanol Yield Based on CO$_2$ Feed** | |
|---|---|---|---|---|
| T [K] | Experimental data [32] | Simulation results | Experimental data [32] | Simulation results |
| 483 | 0.170 | 0.175 | 0.110 | 0.105 |
| 503 | 0.225 | 0.211 | 0.155 | 0.145 |
| 523 | 0.255 | 0.229 | 0.178 | 0.166 |
| 543 | 0.250 | 0.241 | 0.140 | 0.126 |
| SV [mL/g cat·h] | Experimental data [32] | Simulation results | Experimental data [32] | Simulation results |
| 1000 | 0.262 | 0.254 | 0.193 | 0.183 |
| 2000 | 0.260 | 0.248 | 0.191 | 0.176 |
| 4000 | 0.256 | 0.243 | 0.180 | 0.168 |
| 6000 | 0.250 | 0.240 | 0.166 | 0.154 |
| 8000 | 0.243 | 0.235 | 0.153 | 0.151 |
| 10,000 | 0.230 | 0.228 | 0.134 | 0.144 |

### 2.3.2. Parameter Optimization for CO$_2$ Hydrogenation to Methanol

The $CO_2$ hydrogenation to methanol is limited by the thermodynamic equilibrium. Increasing the per-pass conversion of the reactor is beneficial for reducing the cycle volume and energy consumption. In this paper, the impact of the temperature and pressure of the methanol synthesis reactor on the methanol yield is analyzed, as shown in Figure 6. As the temperature rises from 200 to 260 °C, the methanol molar flow rate at the reactor outlet increases and then decreases, at its maximum when the reaction temperature reaches 220 °C. The reactor pressure increases from 30 to 7 bar, and the methanol molar flow rate increases. When the pressure reaches 50 bar, the continued increase in pressure lowers the methanol molar flow rate increase, and a too-high pressure requires a higher intensity of

the reactor. Hence, the temperature and pressure in the reactor are chosen to be 220 °C and 50 bar, respectively.

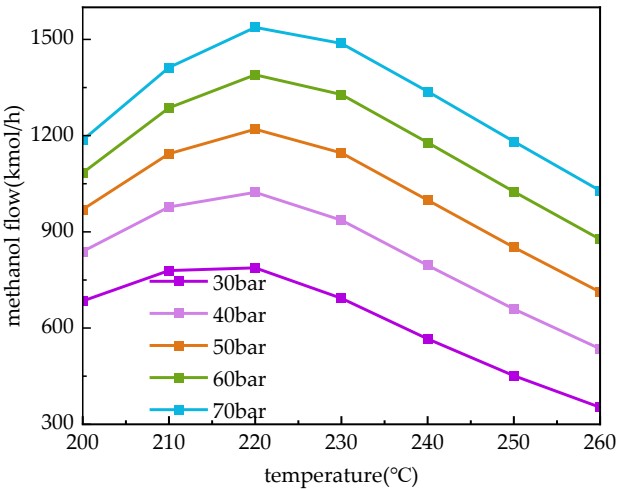

**Figure 6.** Impact of temperature and pressure on the molar flow rate of methanol.

### 2.3.3. Procedure for $CO_2$ Hydrogenation to Methanol

Direct $CO_2$ hydrogenation to methanol is selected in this study, which mainly consists of raw gas compression, methanol synthesis, gas separation, and crude methanol distillation, as shown in Figure 7. The feed $H_2$ is gradually pressurized to 50 bar using a multi-stage compressor with an inter-cooler and mixed with the gas recycled back into the reactor with a Cu/Zn/Al/Zr catalyst [32]. The reaction products are separated by high- and low-pressure flash distillation, the unreacted feed gas is recycled back to the reactor, and the liquid phase is purified via crude methanol distillation for product extraction. Methanol is extracted from the top of the tower, and the bottom of the tower holds wastewater. The reactor for $CO_2$ hydrogenation to methanol is a multi-tube catalytic reactor with a tube length of 12 m, tube diameter of 0.06 m, and a catalyst bed void ratio of 0.5. Due to the low per-pass conversion in the reactor, the reaction products include large amounts of residual $CO_2$, CO, and $H_2$ in addition to the target product methanol. Hence, the gas needs to be separated; most is recycled back, and the remaining liquid is passed to the distillation column for purification. The flow rate results for the $CO_2$ hydrogenation to methanol are shown in the supporting information (illustrated in Table S6).

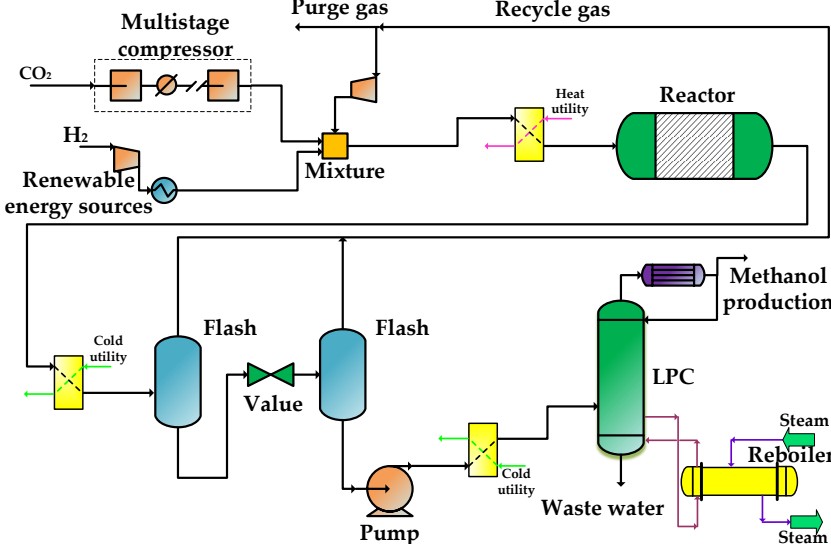

**Figure 7.** Carbon dioxide hydrogenation to methanol.

*2.4. Methanol to Formaldehyde (FS Unit)*

2.4.1. Reaction Mechanism for Producing Methanol from Formaldehyde

　　The main industrial production methods to obtain formaldehyde from methanol are formaldehyde production over an iron-molybdenum oxide catalyst [35] or silver oxide catalyst [36]. The advantages of formaldehyde production over a silver oxide catalyst are a short, mature process, low investment, and low electricity consumption. However, the disadvantages are low methanol conversion, high energy consumption, a short catalyst life, and more methanol residue in the product formaldehyde solution. Compared to the formaldehyde production over a silver oxide catalyst, an iron-molybdenum oxide catalyst has the benefits of a low reaction temperature, low cost, few side reactions, high selectivity, and high yield, especially the high formaldehyde concentration, which is beneficial for reducing the energy consumption of the subsequent glycolic acid separation [37].

　　Through the formaldehyde production over an iron-molybdenum oxide catalyst, methanol reacts with oxygen to produce formaldehyde, and the main reaction is:

$$CH_3OH + 0.5O_2 \rightarrow HCHO + H_2O \tag{20}$$

　　The main byproducts are CO and dimethyl ether (DME). Small amounts of other substances may be produced under different conditions [38], but as these substances are difficult to detect at such very low concentrations, only the higher concentrations of CO and DME are considered, with the following side reactions:

$$HCHO + 0.5O_2 \rightarrow CO + H_2O \tag{21}$$

$$2CH_3OH \rightarrow (CH_3)_2O + H_2O \tag{22}$$

　　The kinetic rate expressions are shown below. The kinetic parameter regression results can be seen in the following Table 5. A comparison of experimental and calculated values for the fraction of each component is shown in Figure S2.

$$r_{HCHO} = \frac{k_1 p_{CH_3OH}}{1 + k_1 p_{CH_3OH}/(k_2 p_{O_2}^{0.5})} \tag{23}$$

$$r_{CO} = \frac{k_3 p_{CH_3OH}^{0.5}}{1 + k_1 p_{CH_3OH}/(k_2 p_{O_2}^{0.5})} \tag{24}$$

$$r_{DME} = \frac{k_4 p_{CH_3OH}^2 - \frac{k_4}{K_{eq}} p_{DME} p_{H_2O}}{1 + k_1 p_{CH_3OH}/(k_2 p_{O_2}^{0.5})} \tag{25}$$

$$\ln(K_{eq}) = -2.2158 + \frac{2606.8}{T} \tag{26}$$

$$k_i = A_{i,0} \exp\left[-\frac{E_i}{R}\left(\frac{1}{T} - \frac{1}{T_0}\right)\right] \tag{27}$$

where $T_0$ is the reference temperature and its value is 573.15 K.

**Table 5.** Kinetic parameter regression results.

| Term | Pre-Exponential Factor $A_{i,0}$/(mol·kg$^{-1}$·s$^{-1}$·MPa$^{-n}$) | Activation Energy $E_i$/(KJ·mol$^{-1}$) |
|------|------|------|
| $k_1$ | 9.928 | 83.42 |
| $k_2$ | 0.8994 | 70.68 |
| $k_3$ | $7.032 \times 10^{-3}$ | 70.42 |
| $k_4$ | 27.55 | 65.95 |

### 2.4.2. Procedure for Producing Methanol from Formaldehyde

In this study, the process of formaldehyde production over iron-molybdenum oxide catalyst is carried out, as shown in Figure 8. The methanol is mixed with air after heat exchange with the reaction products, and the temperature of the mixture is increased to 194 °C. The reaction mixture then enters the multi-tube catalytic reactor to produce formaldehyde and water in the presence of a catalyst. The total reaction is strongly exothermic, and medium pressure steam is generated on the shell side to extract the reaction heat. The reaction product leaves the tube side at 392 °C and enters the pre-heater to exchange heat with the incoming reaction raw material mixture. The reaction product after cooling enters the separator to separate the gas-liquid phase, and the gas phase enters the water wash tower to separate formaldehyde from oxygen, nitrogen, etc., after which it is mixed with the previous liquid phase to obtain the aqueous formalin [39].

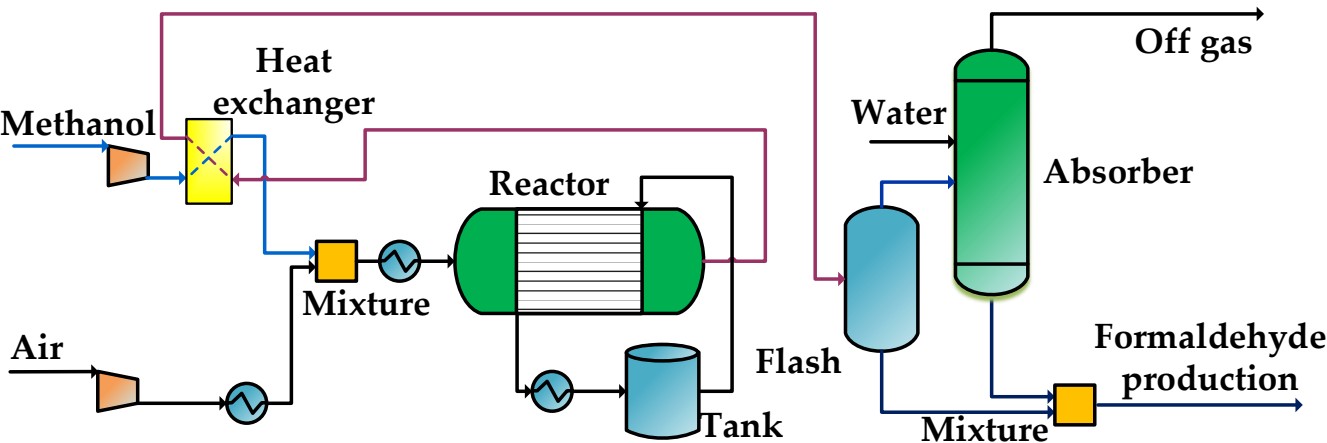

**Figure 8.** Producing methanol from formaldehyde.

The formaldehyde reactor when using an iron-molybdenum oxide catalyst is a multi-tube catalytic reactor with an inlet pressure of 2.16 bar and an inlet temperature of 192 °C. The reactor has a catalyst bed length of 0.77 m, the diameter in the tube is 0.025 m, and the bed void ratio is 0.45.

### 2.4.3. Parameter Optimization for Producing Methanol from Formaldehyde

In methanol oxidation to formaldehyde, the amount of water passing through the water washing column is an important influencing parameter for the formaldehyde concentration and recovery rate. Increasing the amount of water is beneficial for reducing the amount of formaldehyde loss at the top of the column, but it will cause the formaldehyde concentration to increase. The formaldehyde concentration is an important, critical influencing factor for the energy consumption of the next formaldehyde carbonylation step. In this paper, the water feed flow rate of the formaldehyde synthesis water washing column is optimized, as shown in Figure 9.

The water flow rate rises from 8000 to 16,000 kmol/h, and the molar flow rate of formaldehyde loss at the top of the water washing column gradually decreases from 42.8 kmol/h to the minimum value of 15.1 kmol/h. However, the variation is inconspicuous in the range of 8000–10,000 kmol/h, while the formaldehyde concentration at the bottom of the column outlet keeps decreasing, but stays the same after 10,000 kmol/h. Therefore, to meet the requirements of the next step of synthesizing glycolic acid with a minimum flow rate and energy consumption, the water feed flow rate is chosen as 10,000 kmol/h. The flow rate results when transforming methanol to formaldehyde are shown in the supporting information (illustrated in Table S7).

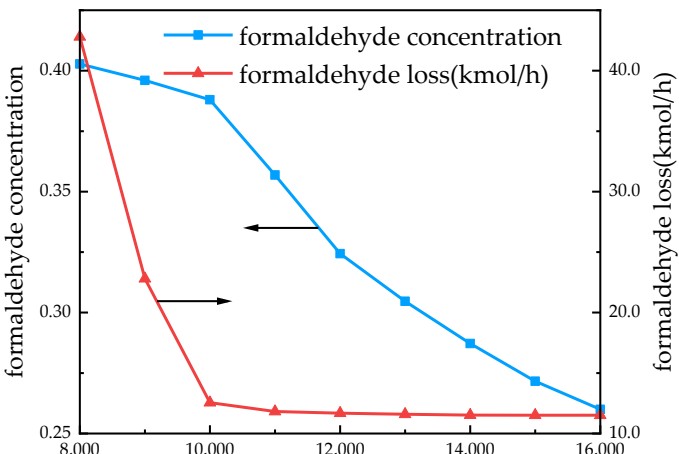

**Figure 9.** Impact of wash tower water intake on formaldehyde concentration and molar flow loss.

### 2.5. Formaldehyde Carbonylation to Glycolic Acid (GAS Unit)

#### 2.5.1. Reaction Mechanism for Formaldehyde Carbonylation to Glycolic Acid

As early as the 1940s, DuPont [40] developed technology for carbonylating formaldehyde to glycolic acid using $H_2SO_4$ as a catalyst, but this method was discontinued in 1968 due to the severe corrosion caused by $H_2SO_4$. To solve the corrosion problem, ionic liquid has been applied in this process. However, there are still disadvantages such as complexity in preparation and difficulty in separation [41]. In comparison, solid acid catalysts [42] such as heteropolyacids, zeolite, and acidic resins are less corrosive, environmentally friendly, and easily recyclable. Thus, much attention has been paid to solid acidic catalysts for formaldehyde carbonylation. Yang et al. successfully developed a green polymetallic solid acid catalyst [43]. It has now been found that it is possible to obtain high glycolic acid yields through the carbonylation of formaldehyde under low temperature and pressure conditions by employing a particular reaction medium and particular catalysts. The catalysts have the advantages of no corrosion or environmental emissions, a low reaction temperature and pressure, high raw material conversion, simple separation and purification, and high product selectivity. The main reactions involved in the carbonylation of formaldehyde to glycolic acid are:

$$HCHO + CO + H_2O \rightarrow HOCH_2COOH \tag{28}$$

$$2HCHO + 2CO + H_2O \rightarrow HOCH_2COOCH_2COOH \tag{29}$$

$$2HOCH_2COOH \rightarrow HOCH_2COOCH_2COOH + H_2O \tag{30}$$

#### 2.5.2. Procedure for Formaldehyde Carbonylation to Glycolic Acid

A fixed bed reactor is applied with a molar ratio of formaldehyde to CO of 1:5. Formaldehyde is mixed with CO and fed through the catalyst bed at a weight hourly space velocity of $2\,h^{-1}$, with a formaldehyde conversion of 97.5% and glycolic acid selectivity of 98.1%. The products are separated at 60 °C in the flash, and the separated liquid phase enters the dehydration tower with a reflux ratio of 0.4 and a tower top product to feed flow rate ratio of 0.5. The process flow is shown in Figure 10. The formalin, carbon monoxide, and mixed gas are heated to 60 °C by a pre-heater and enter the fixed bed reactor. The reaction temperature is controlled at 60 °C and the reaction pressure is 3 bar. The reaction products are separated after cooling, and the incomplete reaction gases are circulated. The liquid phase product (70 wt% glycolic acid solution) enters the distillation column to obtain 81 wt% glycolic acid solution flowing into the crystallizer for crystallization. The liquid-solid separation is then carried out in a centrifugal filter, and the liquid phase continues to be circulated back to the distillation column, while the solid phase is the product.

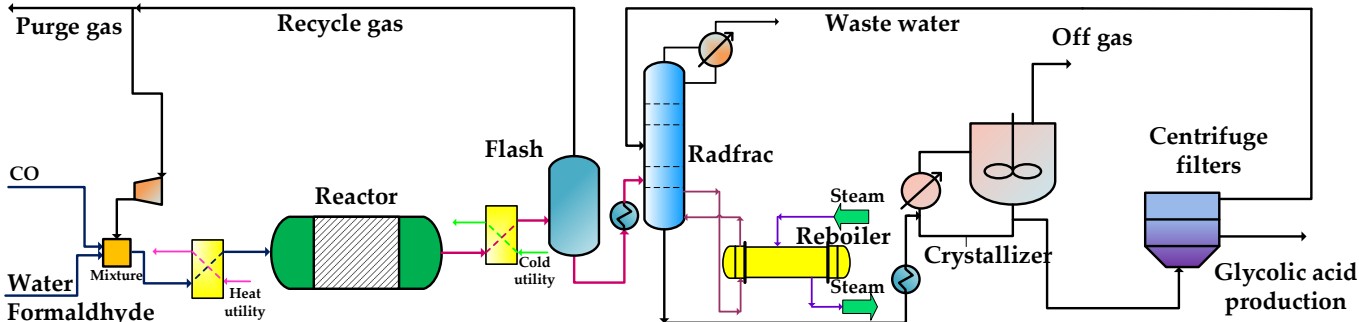

**Figure 10.** Carbonylation of formaldehyde to glycolic acid.

2.5.3. Parameter Optimization for Formaldehyde Carbonylation to Glycolic Acid

The reaction products of formaldehyde carbonylation include carbon monoxide, formaldehyde, water, nitrogen, glycolic acid, and diglycolic acid. The main purpose of the distillation column is to raise the glycolic acid concentration from the flash tank outlet concentration to about 80 wt%, then crystallization is used for further purification. The solubility data for glycolic acid in water are crucial when using the crystallization method to purify glycolic acid, and the solubility model obtained by correlating the solubility data measured by the static equilibrium method is used for the design calculation of the glycolic acid crystallization unit [44]. The solubility data for glycolic acid in water are illustrated in Table S8.

In this paper, the reflux ratio in the distillation column during the synthesis of glycolic acid is optimized, as shown in Figure 11. The water purity at the top of the column increases gradually with the rise in the reflux ratio, then when the reflux ratio exceeds 0.4, the water purity in desorption gas decreases sharply; meanwhile, the duty ascends as the reflux ratio rises. According to the requirements, and considering that more duty causes more energy consumption, the reflux ratio of 0.4 and the duty of 180 MW are chosen to obtain water purity at the top of the column of 0.993.

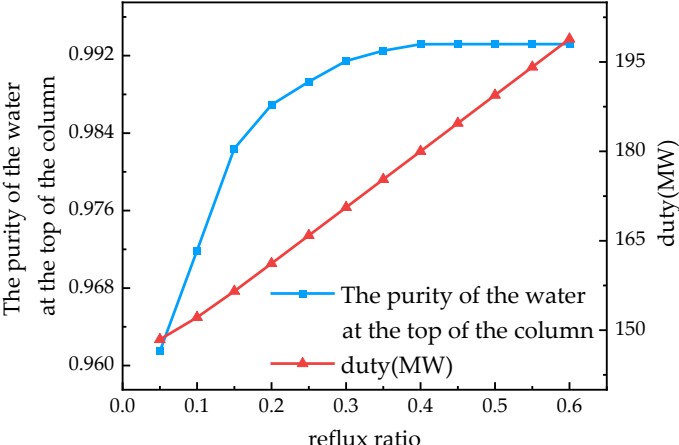

**Figure 11.** Impact of reflux ratio on the water purity at the top of the column and duty.

In this paper, the temperature of the crystallizer is optimized as shown in Figure 12. The yield decreases gradually with the rise in temperature, while the duty increases as the temperature rises. The analysis shows that the crystallization of glycolic acid is a dynamic equilibrium process; the lower the temperature, the lower the solubility of glycolic acid and the more it is crystallized. Considering that the low-temperature operation will cause a rise in energy consumption, it can be seen from Figure 12 that a crystallization endpoint temperature of 5 °C will obtain crystals with 99% purity and 75% yield, which

is just enough to meet the requirements. Therefore, it is appropriate to choose 5 °C as the crystallization endpoint temperature [45].

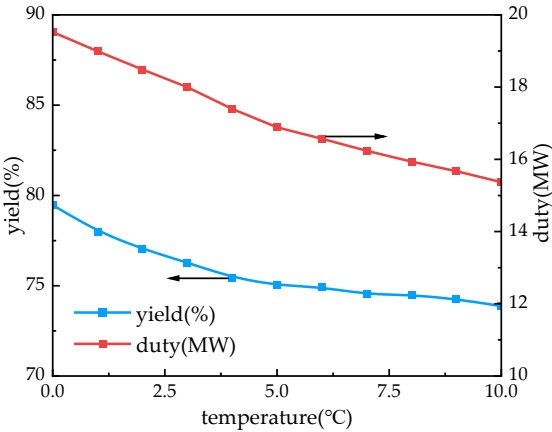

**Figure 12.** Impact of temperature in the crystallizer on yield and duty.

The flow rate results for formaldehyde carbonylation to glycolic acid are shown in the supporting information (illustrated in Table S9).

### 2.6. Comparison of the Simulation Results and Experimental Data

Based on the above optimization conditions, a comparison of the simulation results and experimental data is presented in Table 6.

**Table 6.** Comparison of the simulation results and experimental data.

| Unit | Key Parameters | Simulation Results | Experimental Data | Ref. |
|---|---|---|---|---|
| CC unit | Upper phase volume fraction (%) | 41.20 | 43.60 | [28] |
| | Reboiler duty, GJ/t-$CO_2$ | 2.41 | 2.40 | |
| | 1-propanol loss ratio (%) | 2.56 | 2.42 | |
| MS unit | Methanol composition at reactor outlet (wt%) | 12.40 | 12.00 | [32] |
| | Top mass flow rate of the distillation column, t/h | 55.40 | 55.10 | |
| | Mass fraction of methanol, % | 99.90 | 99.96 | |
| | Reactor outlet flow rate, t/h | 472.80 | 467.60 | |
| FS unit | Methanol conversion (%) | 99.00 | 98.80 | [37] |
| | Selectivity of formaldehyde (%) | 88.20 | 85.75 | |
| | Yield of formaldehyde (%) | 96.23 | 93.33 | |
| GAS unit | HCHO conversion | 0.975 | 0.961 | [42] |
| | Glycolic acid selectivity | 0.957 | 0.943 | |

## 3. Energy Utilization and Analysis

To improve the energy utilization and reduce the operating costs, heat integration is carried out based on the pinch point analysis method, and grand composite curves (GCCs) of the $CO_2$ capture, $CO_2$ hydrogenation to methanol, methanol oxidation to formaldehyde, and formaldehyde carbonylation are plotted, as shown in Figure 13. Temperature parameters for utilities are illustrated in Table S10.

The utilities required for the carbon capture include medium-pressure steam and circulating cooling water. The medium-pressure steam is used for the reboiler of the stripper with a load of 166.6 MW. $CO_2$ hydrogenation to methanol mainly uses low- and high-pressure steam. Low-pressure steam with a load of 43.2 MW is used for the reboiler of the distillation tower. High-pressure steam is mainly used for heating reaction raw materials, and the load is 13.0 MW. Transforming methanol to formaldehyde is an

exothermic process, which releases a large amount of heat. A total heat of 207.0 MW can be recovered, including 5.5 MW heat from a large amount of medium-pressure steam and 201.5 MW heat from a small amount of medium-pressure steam. Formaldehyde carbonylation to glycolic acid mainly uses two grades of low- and medium-pressure steam, both of which are used in the distillation column of the reboiler. The load of low-pressure steam is 138.8 MW, and the load of medium-pressure steam is 48.0 MW. Besides this, the refrigerant reduces some of the energy in the crystallizer to the desired low temperature. The remaining unrecoverable heat is cooled by circulating cooling water.

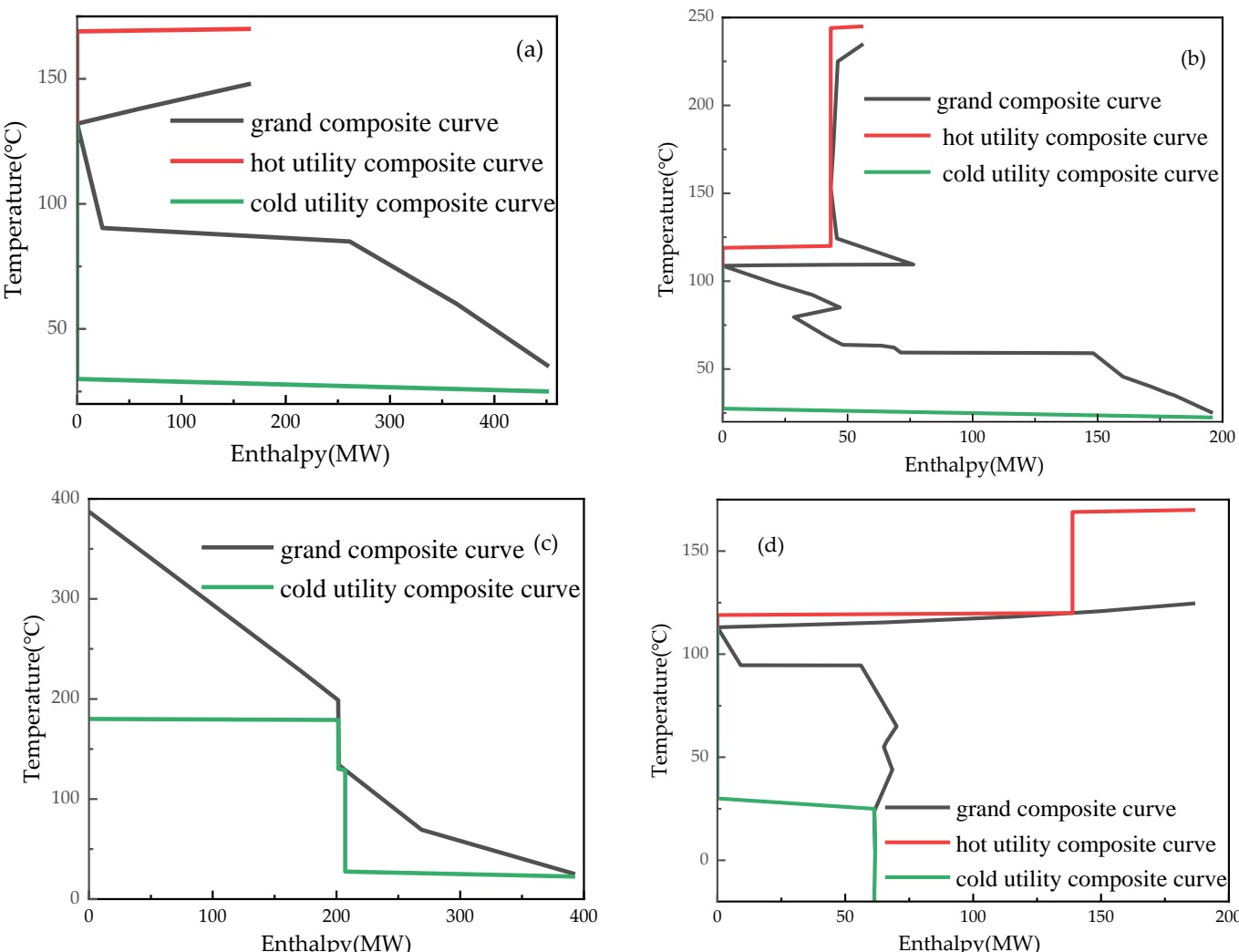

**Figure 13.** Grand composite curve for each unit ((**a**) CC unit; (**b**) MS unit; (**c**) FS unit; (**d**) GAS unit).

To improve the energy utilization of the entire process system, this study establishes whole-process heat integration with utility coupling heat exchanger networks from the perspective of energy allocation of each unit and utilization of waste heat, to match the flow units within the plant and realize the supply and demand distribution of different levels of steam between units. Figure 14 shows the total heat source available for all contributing units (red curve on the left side of the figure) and the total heat sink that necessarily provides heat from an external heat source (black curve on the right side of the figure).

As can be seen from Figure 14, the heat released by the total system is 1218.2 MW, while the heat required by the system is only 415.8 MW. The heat released by the process system is much higher than the heat required. The main reason is massive exothermic reactions in the process. As can be seen from Figure 14a, the total heat source curve and the total heat

sink curve do not intersect, which indicates that the heat section is not fully utilized, and the process system has the potential to improve the recovery of part of the steam. The use of steam can be reduced by thermal integration between units. As shown in Figure 14b, the heat generated by transforming methanol to formaldehyde can be provided to other units in the process, including the medium-pressure steam required for carbon capture and formaldehyde carbonylation to glycolic acid, and the low-pressure steam required for $CO_2$ hydrogenation to methanol and formaldehyde carbonylation to glycolic acid.

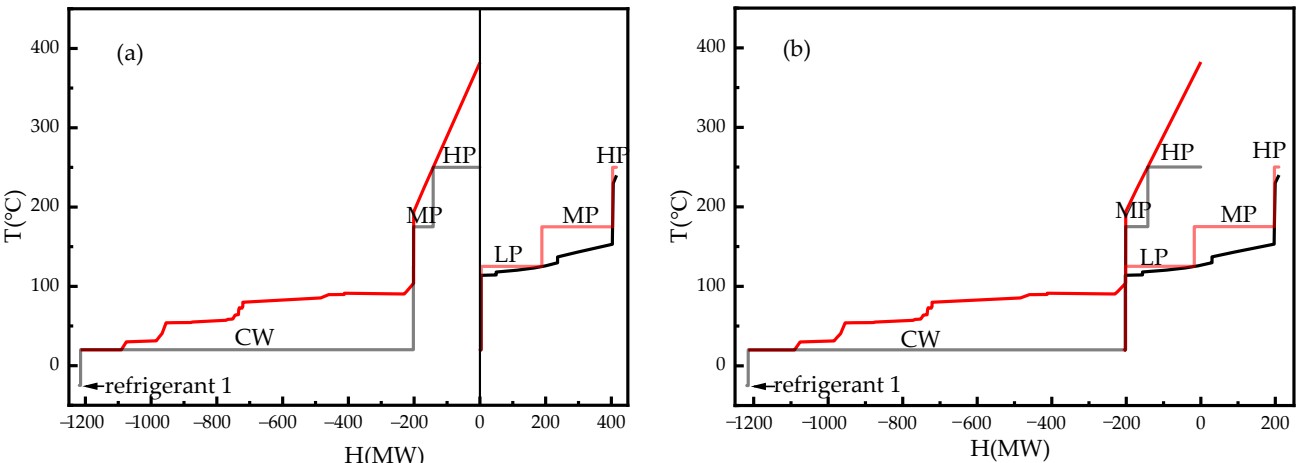

**Figure 14.** Total site heat integration ((**a**) before total site heat integration; (**b**) after total site heat integration).

The energy consumption distribution of the four units is shown in Figure 15a. The energy consumption of the $CO_2$ capture unit includes 452.3 MW of cooling water and 166.6 MW of medium-pressure steam. The energy consumption of the formaldehyde carbonylation to glycolic acid unit is 61.33 MW of cooling water, 0.36 MW of refrigerant, 138.8 MW of low-pressure steam, 47.98 MW of medium-pressure steam, and 4.57 MW of electric power consumption. The energy consumption of the methanol oxidation to formaldehyde unit is 452.3 MW of cooling water, producing 201.5 MW of medium-pressure steam and 5.53 MW of low-pressure steam. The energy consumption of the formaldehyde carbonylation to glycolic acid unit is 61.33 MW of cooling water, 0.36 MW of refrigerant, 138.8 MW of low-pressure steam, and 47.98 MW of medium-pressure steam, along with 4.57 MW of electricity consumption.

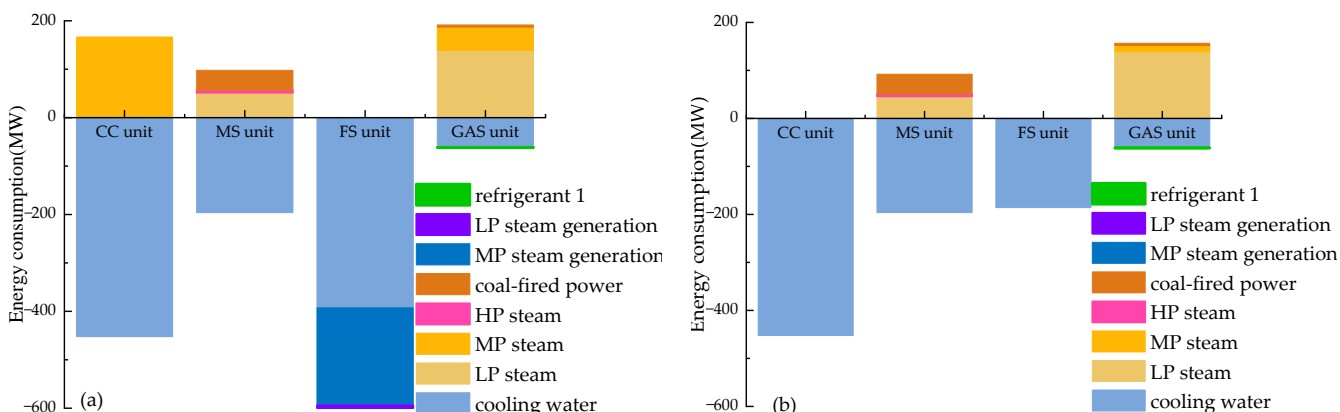

**Figure 15.** Energy consumption distribution without or with total site heat integration ((**a**) energy consumption before total site heat integration; (**b**) energy consumption after total site heat integration).

After total energy integration, this optimal result reduces the total cost by 27.4%. As shown in Figure 15b, the energy consumption of the $CO_2$ capture unit is only 452.3 MW

of cooling water; the energy consumption of the $CO_2$ hydrogenation to methanol unit is 196.1 MW of cooling water, 46.16 MW of low-pressure steam, 4.6 MW of high-pressure steam, and 41.41 MW of electricity consumption; the energy consumption of the methanol oxidation to formaldehyde unit is 185.8 MW of cooling water; the formaldehyde carbonylation to glycolic acid unit's energy consumption is 61.33 MW of cooling water, 0.36 MW of refrigerant, 138.8 MW of low-pressure steam, 13.08 MW of medium-pressure steam, and 4.57 MW of electric power consumption.

## 4. Results and Discussion

### 4.1. Process Simulation Results

Based on the above optimization conditions, the key parameters for each section are shown in Table 7. Under these conditions, strict process modeling and simulation are carried out for the whole process, and the key flow rate results are shown in Table 8.

**Table 7.** Key parameters for process modeling and simulation.

| Unit | Key Parameters | Value | Key Parameters | Value |
|---|---|---|---|---|
| CC unit | Absorber temperature | 42 °C | Stripper temperature | 120 °C |
| | MEA consumption | 46 t/h | $C_3H_8O$ consumption | 252 t/h |
| | Upper phase volume fraction | 41.2% | $CO_2$ capture rate | 90% |
| MS unit | Methanol reactor temperature | 220 °C | Methanol reactor pressure | 5 bar |
| | $CH_4O$ mass fraction | 99.7% | Reflux ratio | 0.9 |
| FS unit | Catalyst bed length | 0.77 m | Tube diameter | $2.5 \times 10^{-2}$ m |
| | Inlet pressure | 2.16 bar | Inlet temperature | 194 °C |
| | Methanol conversion | 96.1% | Bed porosity | 0.5 |
| | Catalyst utilization time | 12–18 months | Particle density | 1000 kg/cum |
| GAS unit | Glycolic acid reactor temperature | 60 °C | Glycolic acid reactor pressure | 3 bar |
| | Reaction space velocity | 2 $h^{-1}$ | Formaldehyde conversion | 97.5% |
| | Glycolic acid selectivity | 98.1% | Ratio of formaldehyde to carbon monoxide | 0.2 |

**Table 8.** Key flow rate results for the whole process.

| Stream | 1 | 2 | 3 | 4 | 5 | 6 | 7 |
|---|---|---|---|---|---|---|---|
| Temperature (°C) | 86.00 | 30.00 | 42.00 | 40.00 | 63.57 | 32.93 | 5.00 |
| Pressure (bar) | 50.00 | 3.00 | 1.00 | 1.00 | 1.00 | 5.00 | 1.10 |
| Molar flow rate (kmol/h) | | | | | | | |
| $H_2O$ | 0.00 | 0.00 | 1680 | 62.333 | 7.44 | 13,165.39 | 0.93 |
| $H_2$ | 16,387.13 | 0.00 | 0.00 | 0.00 | 0.00 | 0.00 | 0.00 |
| CO | 0.00 | 4945.63 | 0.00 | 0.00 | 0.00 | 0.00 | 0.00 |
| $N_2$ | 0.00 | 0.00 | 31,160 | 0.00 | 0.00 | 0.39 | 0.00 |
| $O_2$ | 0.00 | 0.00 | 1320 | 0.00 | 0.00 | 0.30 | 0.00 |
| $CO_2$ | 0.00 | 0.00 | 5840 | 5256.74 | 8.41 | 0.00 | 0.00 |
| $C_3H_8O$ | 0.00 | 0.00 | 0.00 | 16.268 | 0.00 | 0.00 | 0.00 |
| $CH_4O$ | 0.00 | 0.00 | 0.00 | 0.00 | 5139.41 | 1.77 | 0.00 |
| HCHO | 0.00 | 0.00 | 0.00 | 0.00 | 0.00 | 4945.63 | 0.00 |
| $C_2H_6O$ | 0.00 | 0.00 | 0.00 | 0.00 | 0.00 | 0.00 | 0.00 |
| $OHCH_2COOH$ | 0.00 | 0.00 | 0.00 | 0.00 | 0.00 | 0.00 | 0.25 |
| Diglycolic acid | 0.00 | 0.00 | 0.00 | 0.00 | 0.00 | 0.00 | 0.01 |
| $C_2H_4O_3(S)$ | 0.00 | 0.00 | 0.00 | 0.00 | 0.00 | 0.00 | 4818.99 |

### 4.2. Carbon Utilization Rate

The carbon utilization rate does not only affect the efficiency of resource utilization and economic efficiency but also has an impact on the cost and $CO_2$ emissions in the production

process [46]. The carbon utilization rate is defined as shown in the below equation, which is the molar ratio of carbon in the product output to carbon in the raw material input.

$$E_{CC} = \frac{\sum n_{C,products}^{out}}{\sum n_{C,feed}^{in}} \times 100\% \tag{31}$$

The process of carbon flow is shown in Figure 16, where the feed carbon molar flow rate is 5840 kmol/h, the carbon molar flow rate in the product glycolic acid is 4818.99 kmol/h, and the carbon utilization rate is 82.5%. The direct $CO_2$ emissions are 583.26 kmol/h from the tail gas in $CO_2$ capture, 117.33 kmol/h from the purge gas in $O_2$ hydrogenation to methanol, 193.78 kmol/h from the emission gas of the water washing column in methanol oxidation to formaldehyde, and 126.64 kmol/h from the purge gas in the formaldehyde carbonylation to glycolic acid.

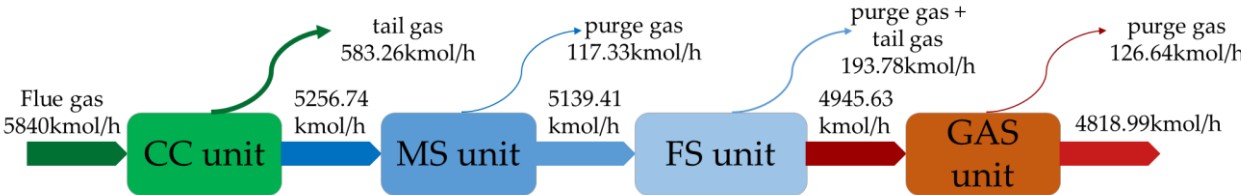

**Figure 16.** Process of carbon flow and rate.

### 4.3. $CO_2$ Emissions

The direct [47] and indirect [48] $CO_2$ emissions are considered in this study as formulated in the following equation. Direct emissions represent the greenhouse gas $CO_2$ emissions of this process, which can be calculated from the process of carbon flow in Figure 16. Meanwhile, the indirect emissions from utilities (steam, electricity, etc.) cannot be ignored [49]. The energy consumption and emission factors are illustrated in Table S11.

$$\eta_e = \frac{E_{CO_2}^d + E_{CO_2}^{ind}}{m_{GA}} = \frac{E_{CO_2}^d + \sum EC_j \times p_{j,CO_2}}{m_{GA}} \tag{32}$$

where $E_{CO_2}^d$ and $E_{CO_2}^{ind}$ are direct and indirect $CO_2$ emissions, $p_{j,CO_2}$ represents the indirect $CO_2$ emissions from $j$th utility's consumption, and $EC_j$ represents the consumption of the $j$th utility.

Steam is assumed to be generated by a natural gas-fueled boiler with an assumed combustion efficiency of 80% [50]. All the electricity required for the process comes from thermal power generation, and the $CO_2$ emissions of the whole process are 2913.32 kmol/h, with a $CO_2$ emission ratio of 0.35 kg-$CO_2$/kg-GA.

The direct and indirect $CO_2$ emissions of the four units are calculated and analyzed, and the results are shown in Figure 17. Among the direct emissions, the CC unit has the largest proportion of carbon emissions. It is necessary to further screen the absorbents to select one with a low heat of desorption reaction, and develop a new desorption process to reduce energy consumption. The main reason is the utility energy consumption of the feed $H_2$ multi-stage compression and crude methanol distillation. Meanwhile, when considering the indirect emissions of each unit, the MS unit had the largest ratio of carbon emissions, followed by the GAS unit. The main reason for MS unit, again, is the utility energy consumption of the feed $H_2$ multi-stage compression and crude methanol distillation. In terms of the GAS unit, it is mainly consumed by distillation and concentration of the reaction product, crude glycolic acid. Therefore, the bottlenecks in $CO_2$ indirect emissions are the $CO_2$ hydrogenation to methanol and the formaldehyde carbonylation to glycolic acid, which need to be studied further to determine the energy-saving potential and optimize the energy consumption, to reduce the indirect emissions from the two units.

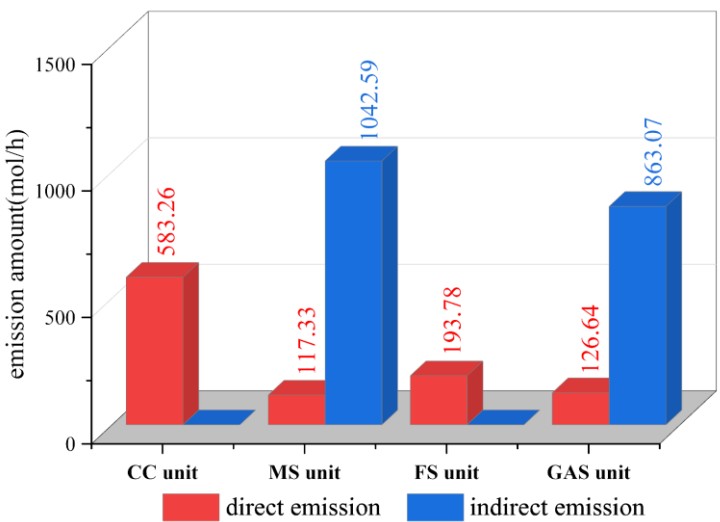

**Figure 17.** Direct and indirect emissions from the unit.

*4.4. Techno-Economic Analysis*

　　Total production cost (TPC) estimation is one of the most important bases for the project decisions made for a newly designed process. It is the sum of the equipment investment (EI, k$/year) and total operating cost (TOC, k$/year). The former mainly includes the costs of all production equipment and their related facilities; the latter mainly includes the costs required to maintain the normal production operation of the enterprise [51]. The unit production cost (UPC, $/kg) is also given to further assess the process economics. The calculation process is based on the production of 4818.99 kmol/h of glycolic acid and an annual running time of 8000 h/year. Input parameters and assumptions for techno-economic analyses are illustrated in Table S12.

$$TCI_i = \frac{EI_i}{\text{period}} + TOC_i \tag{33}$$

$$UPC_i = \frac{TCI_i}{m_{\text{GA}}} \tag{34}$$

where $TPC_i$ is the total production cost required for the $i_{\text{th}}$ unit; $EI_i$ is the equipment investment required for the $i$th unit; $TOC_i$ is the total operation cost required for the $i$th unit; $UPC_i$ is the unit production cost of $i$th unit; $m_{\text{GA}}$ is the mass flow rate of glycolic acid. Based on the above calculation benchmarks and formulas, the results of the economic analysis are shown in Table 9.

**Table 9.** Comparative economic results.

| Item | EI, k$/Year | TOC, k$/Year | TPC, k$/Year | Proportion, % | UPC, $/kg | Product |
|------|-------------|--------------|--------------|---------------|-----------|---------|
| CC | 42,077.00 | 67,967.35 | 73,975.94 | 5.22 | 39.98 | $CO_2$ |
| MS | 114,889.30 | 887,673.10 | 1,002,562.40 | 70.80 | 595.08 | $CH_3OH$ |
| FS | 24,788.20 | 34,670.88 | 59,459.08 | 4.20 | 652.55 | HCHO (solution) |
| GAS | 89,243.20 | 190,739.19 | 279,982.39 | 19.77 | 834.75 | $HOCH_2COOH$ |
| Total | 27,0997.70 | 1,181,050.51 | 1,415,979.81 | | 834.75 | |

　　According to Table 9, the unit production cost of the process is 834.75 $/t-GA; the $CO_2$ hydrogenation to methanol accounts for the largest proportion of the total production cost, which is 70.8%. The total production cost of the unit is mainly limited by the price of hydrogen (3.13 $/kg), which has an impact on the TOC. As the efficiency of hydrogen production by water electrolysis improves and the investment cost of electrolyzer decreases in the future, when the electricity price falls to 0.02–0.03 $/kWh and

the price of $H_2$ is 1.17–1.46 \$/kg, the total production cost of carbon dioxide hydrogenation to methanol will be 48,5043–56,0428 k\$/year. The total production cost will then be 898,460.71–974,599.71 k\$-year$^{-1}$, and the investment cost of this process will be further reduced to 529.7–574.1 \$/kg.

## 5. Conclusions

In this paper, a novel process for synthesizing glycolic acid based on $CO_2$ capture coupling green hydrogen was proposed, which includes power-to-hydrogen, $CO_2$ capture, $CO_2$ hydrogenation to methanol, methanol oxidation to formaldehyde, and formaldehyde carbonylation to glycolic acid. The technical, economic, and environmental performance were evaluated based on the optimal key operational parameters. The main results gained from this work can be summarized as follows:

(1) The carbon utilization rate can reach 82.5%, and the $CO_2$ emissions are then 0.35 kg-$CO_2$/kg-GA. Among the direct emissions of each unit, the CC unit emits the largest proportion of carbon. Meanwhile, when considering the indirect emissions of each unit, the MS unit emits the largest ratio of carbon, followed by the GAS unit;

(2) This study establishes total site energy integration based on the pinch point analysis method to improve the energy utilization and reduce the operating costs. After total site energy integration, the optimal result has a better comprehensive performance, which reduces the total consumption by 27.4%;

(3) The unit production cost of the proposed process is 834.75 \$/t-GA. Due to the high green hydrogen price, $CO_2$ hydrogenation to methanol accounts for the largest proportion, accounting for 70.8% of the total production cost. Fortunately, with the rapid development of renewable energy generation and power-to-hydrogen technologies, the renewable $H_2$ price will continue to drop, and the production cost of the proposed process will be further reduced.

$CO_2$ resource utilization into glycolic acid provides a new solution for greenhouse gas reduction. We hope the models and results obtained in this study can be used to guide the production of high-value products from $CO_2$ and renewable $H_2$.

**Supplementary Materials:** The following supporting information can be downloaded at: https://www.mdpi.com/article/10.3390/pr10081610/s1, Figure S1. Volume fraction of the upper phase and $CO_2$ distribution in the 30 wt% MEA/40 wt% 1-propanol biphasic solvent at different time [52]; Table S1. Comparison of the experimental density and the model density of MEA/1-propanol system with different $CO_2$ loading (at 293 K); Table S2. The flow rate results of the unit of carbon capture; Table S3. Reaction kinetic parameter of k1, k2 and k3; Table S4. Constants for driving force and chemical equilibrium data; Table S5. Ki factors for adsorption term; rest is explicitly derived by calculation) [32,33]; Table S6. The flow rate results of the unit of $CO_2$ hydrogenation to methanol; Figure S2. Experimental and calculated values for the fraction of each component [53]; Table S7. The flow rate results of the unit of methanol to formaldehyde; Table S8. Solubility of glycolic acid in water; Table S9. The flow rate results of the unit of formaldehyde carbonylation to glycolic acid; Table S10. Temperature parameters for utilities; Table S11. The parameters of indirect $CO_2$ emission; Table S12. Input parameters and assumptions for techno-economic analyses.

**Author Contributions:** Conceptualization, D.W. and J.L.; methodology, D.W.; software, H.Z., Y.Y. and Z.F.; validation, W.M., J.W. and K.W.; formal analysis, D.W. and J.L.; investigation, D.W. and J.L.; resources, D.W. and J.L.; writing—original draft preparation, J.L.; writing—review and editing, D.W.; supervision, D.W.; project administration, X.F.; funding acquisition, D.W. All authors have read and agreed to the published version of the manuscript.

**Funding:** This research was funded by the Higher Education Industry Support plan project of Gansu Province (no. 2020C-06) and the Young Doctor Fund of the Education Department of Gansu Province (no. 2022QB-214).

**Institutional Review Board Statement:** Not applicable.

**Informed Consent Statement:** Not applicable.

**Data Availability Statement:** The data presented in this study are available on request from the first author.

**Conflicts of Interest:** The authors declare no conflict of interest.

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
