# Peer review of "Integrated Process for Producing Glycolic Acid from Carbon Dioxide Capture Coupling Green Hydrogen"

_processes, doi:10.3390/pr10081610_

Round 1

Reviewer 1 Report

The authors report a proposed process design for glycolic acid production from carbon dioxide coupling with green hydrogen. The manuscript is significant for current situation regarding to the concern of greenhouse gas effect. There is only  a minor revision before publication. As scientific results obtained in this work follow the calculations as they are; however, it could not be ensure that the calculations are correct. Could the authors confirm? In addition, discussion and comparison with literature would be worthwhile.

Reviewer 2 Report

                                            Review on the paper

«Integrated process for glycolic acid from carbon dioxide coupling green hydrogen»

by Dongliang Wang, Jingwei Li, Wenliang Meng, Jian Wang,

Ke Wang, Huairong Zhou, Yong Yang and Zongliang Fan, and Xueying Fan

This paper is devoted to the integrated process of production of glycolic acid (GA) from CO2. This process consists of the following main processes (blocks):

(1) CO2 capture; (2) CO2 hydrogenation to methanol;

(3) methanol oxidation to formaldehyde, and (4) formaldehyde carbonylation processes.

The subject is extremely topical, and the structure of the paper is well-defined; therefore, I recommend publishing this paper.

However, I found the serious drawbacks.

1.     The paper is written as a short story (‘digest’). As a reader, I am interested in reasoning based on the solid physico-chemical basis and corresponding theoretical/mathematical model

 Every process (block) must be supplemented by the explanation of main trends of process characteristics regarding the parameters. It must be shown that this model is used as a basis for choosing the optimal apparatus/reactor and theoretical optimization of the regime.

Examples: CO2 capture. Parameters of the regime (composition of the reactive mixture, molar flow rate, temperature, and pressure) are presented in Table 2.  It is written that the kinetic model is explained in the supplementary materials.

However, what is a link between this kinetic model and trends of the process? By the way, trends are more important than the concrete values of parameters.

CO2 hydrogenation to methanol.

Langmuir-Hinshelwood model is used. However, what are main conclusions and trends based on this model.? Three overall reactions of this process are considered. Two reactions are exothermic, one is endothermic. What are technological consequences of this feature? By the way, the reversibility of these reactions is not indicated.

Similar comments can be formulated for methanol oxidation to formaldehyde,

and

formaldehyde carbonylation process.

I recommend authors to write two-three corresponding paragraphs for every process and answer my questions.

 2.     Page 10. “Increasing the amount of water is beneficial to reducing the amount of formaldehyde

loss at the top of the column, but it will cause formaldehyde concentration.”

What does it mean “it will cause formaldehyde concentration”?

3. In Section 3 “Energy utilization and analysis”, Figures 13 and 14 are not explained well.

My recommendation is “major revision”.

Reviewer 3 Report

The authors propose a study for capture and conversion of the carbon dioxide emitted from coal-fired power plants, which provides a path to feasible low-carbon and clean use of carbon dioxide resources. The results of this research show a discussion about bottlenecks from the perspective of carbon emissions and economic analysis.

The study is worth to be investigated, it is reasonably motivated and addressed using a suitable methodology. The manuscript is well written in general, particularly, the introduction is comprehensive, including an extensive and up-to-date list of references.

In summary, in my opinion, the work deserves to be considered for publication, but a few things must be improved.

I suggest including at the end of the introduction a paragraph with a description of each section of this investigation.

In Section 2, I recommend a brief explication of this section. Also, a description of the optimization model used can be explained.

The column values of all Tables should be aligned to the right and with the same decimal places.

I recommend rewriting the conclusion section by using better words that show to the reader the conclusion of the results of this research.

Round 2

Reviewer 2 Report

I am satisfied by revision